# Factors related to willingness to participate in biomedical research on neglected tropical diseases: A systematic review

Vinícius Raimundo-Silva[1], Caio Torres Marques[1], João Rezende Fonseca[1], Martha Silvia Martínez-Silveira[2], Mitermayer Galvão Reis[1,2,3]*

1 Faculty of Medicine of Bahia, Federal University of Bahia, UFBA, Salvador, Bahia, Brazil, 2 Gonçalo Moniz Institute, Oswald Cruz Foundation, Ministry of Health, Salvador, Bahia, Brazil, 3 Department of Epidemiology of Microbial Diseases, School of Public Health, Yale University—New Haven, Connecticut, United States of America

* mitergreis@gmail.com

**Data Availability Statement:** All relevant data are within the manuscript and its Supporting Information files.

## Abstract

### Background

Understanding the barriers to and facilitators of participation in research could enhance recruitment rates for biomedical research on Neglected Tropical Diseases (NTDs) and help to avoid the problems associated with poor recruitment.

### Methodology/principal findings

We conducted a systematic review to identify factors related to willingness to participate in biomedical research on Neglected Tropical Diseases (NTDs). Our search included the following databases: Medline/PubMed, Embase (Embase.com), Global Index Medicus (WHO), Web of Science (Core collection), and gray literature. We included studies that analyzed or reported factors associated with willingness to participate in NTD research, using either quantitative methods (such as clinical trials, cohorts, and cross-sectional studies) or qualitative methods (such as focus group discussions, semi-structured interviews, and in-depth interviews). There were no language restrictions, but we excluded review articles, notes, case reports, letters to the editor, editor's notes, extended abstracts, proceedings, patents, editorials, and other editorial materials.

Screening of citations, data extraction, and risk of bias assessment was conducted by independent reviewers, according to the study protocol registered on PROSPERO. For analyses, we assessed the frequency of barriers, enablers, and the frequency of recruitment interventions mentioned in the included studies. The protocol for this systematic review was registered under registration number CRD42020212536. (S1 Appendix)

We identified 2070 citations, 1470 from the databases, and 600 from other sources. From those, eleven studies were selected for data extraction and analysis. The studies were conducted in Africa, Asia, and North America. Personal health benefits, monetary benefits, and community engagement and sensitization strategies were identified as the main reasons for participating in biomedical research on Neglected Tropical Diseases (NTDs).

**Funding:** This study was funded by National Institutes of Health (NIH), Grant/Award Numbers: R01 AI121330 to MGR, FAPESB-PNX0001/2015 to MGR for the submitted work, and partially supported by Conselho Nacional de Desenvolvimento Científico e Tecnológico – CNPq (Proc. Proc. 306464/2016-0 and 423833/2018-9) to VRS. VRS is currently receiving a scholarship from Conselho Nacional de Desenvolvimento Científico e Tecnológico (CNPq). The funders had no role in study design, data collection and analysis, decision to publish, or preparation of the manuscript.

**Competing interests:** The authors have declared that no competing interests exist.

However, distrust in researchers, lack of knowledge about research methods among potential participants, and previous negative experiences were identified as the main barriers to participating in biomedical research on NTDs.

## Conclusions/significance

This systematic review provides recommendations for improving adherence to biomedical research on Neglected Tropical Diseases, which can be applied in practice.

### Author summary

Neglected Tropical Diseases (NTDs) are a group of around 25 health issues caused by different germs, and they bring about serious health, social, and economic problems. Those diseases mainly affect poor communities in tropical areas, and some are found in other regions too. Despite their big impact, NTDs don't get much attention or funding from global agencies.

Studying NTDs is crucial to gather information that can help control and eliminate those diseases. However, there are challenges in doing this research, like having a hard time convincing enough people to participate. This is important to fix because if not enough people take part, the research results may not be reliable, and it can make the study take longer and cost more.

We did a careful review to understand what helps and what makes it challenging to convince people to participate in biomedical research on NTDs. Our findings can be useful for researchers to plan better, improve how many people take part in NTD studies, and avoid problems that come with not having enough participants.

## Introduction

Since recruitment is an essential step in biomedical research involving human subjects, poor recruitment increases the possibility of the study being underpowered and introducing selection bias, which can lead to the overgeneralization of results [1,2]. Most studies fail to achieve their original recruitment rate, and around half of them need to extend the enrollment period. One of the main solutions to address poor recruitment is to extend the length of the research. However, this approach increases the cost and workload of the research team [3,4]. Researchers usually have to manage limited resources for their research, and having knowledge about facilitators and barriers of research, especially during the recruitment stage, can be helpful for better planning and overcoming challenges. In the past two decades, numerous studies have examined recruitment in various contexts [3,5–7], and have reported on barriers, motivations, and recruitment strategies that vary based on the research setting, methodology, and target population [8].

The World Health Organization (WHO) defines Neglected Tropical Diseases (NTDs) as a diverse group of communicable diseases that are prevalent in tropical and subtropical regions, affecting approximately one billion people and costing developing economies billions of dollars every year [9]. As of June 25th, 2020, WHO listed 20 diseases in its NTD portfolio [10]. Considering that NTDs are more prevalent in socioeconomically vulnerable communities, social determinants such as poverty, limited healthcare services, and illiteracy may limit the efficacy of recruitment [11].

In order to address the social and financial impacts caused by NTDs, public and private institutions have funded interventions aimed at controlling, eliminating, and eradicating those diseases [12]. Research plays a crucial role in providing evidence to improve interventions for preventing and treating NTDs. Research to identify etiological agents, disease management, and cost-benefit analysis is essential for defining what interventions should be implemented and what the likely impact would be in the contexts where they will be applied [13]. However, biomedical research involving human subjects, especially in the context of scarce resources, faces many barriers [5].

Research on NTDs has been underfunded, with only $100 million allocated in the United States in 2016, compared to $1.5 billion allocated to research on other diseases such as malaria, human immunodeficiency virus, and tuberculosis combined [14]. Since the recruitment problem directly impacts the research results and considering the scarce resources, it is necessary to study and better understand the reasons for this problem and the possible solutions. However, we did not find any systematic review that focuses specifically on the reasons why people refuse to participate in biomedical NTD research. Therefore, we conducted a systematic review to increase the understanding of the reasons for and against participating in biomedical research on NTD.

## Methods

The protocol for this systematic review was registered in the PROSPERO database under registration number CRD42020212536. (S1 Appendix)

The Preferred Reporting Items for Systematic Reviews and Meta-Analyses (PRISMA) were followed for reporting systematic reviews in this article. (S2 Appendix)

### Search strategy and selection criteria

The inclusion criteria for this systematic review were studies that reported factors associated with willingness to participate in biomedical research on NTDs. The types of studies included were quantitative studies (e.g., clinical trials, cohorts, cross-sectional studies) and qualitative studies (e.g., focus group discussions, semi-structured interviews, and in-depth interviews). There were no language restrictions, but reviews, notes, case reports, letters to the editor, editor's notes, extended abstracts, proceedings, patents, editorials, and other editorial materials were excluded. There was no restriction on the publication period of scientific publications.

The search strategy was based on four main steps that involved combining terms and subject headings and managing resources for the search strategy according to each database. The first step aimed to recover biomedical studies and other study designs for clinical research, while the second step recovered studies reporting community participation in research. The third step focused on studies reporting willingness or unwillingness to participate in research, and the fourth step aimed to recover studies on NTD, for which all terms related to the list of Neglected Tropical Diseases defined by the World Health Organization were included. In the final stage of the search strategy, the four steps were combined to obtain the final result. No limits or filter were used.

The searches in database were conducted by two researchers (MMS and VRS). We have chosen to utilize the following electronic databases: Medline via PubMed, Embase via Embase. com by Elsevier, Web of Science Core Collection via Clarivate Analytics and Global Index Medicus via World Health Organization (WHO) which is a platform that allows simultaneous searches in the following databases: African Index Medicus (AIM), Latin American and the Caribbean Literature on Health Science (LILACS), Western Pacific Region Index Medicus (WPRIM), Index Medicus for the Eastern Mediterranean Region (IMEMR), Index Medicus

for the South-East Asian Region (IMSEAR). The PubMed search was conducted in July 2020, while the Embase, Global Index Medicus, and Web of Science searches were conducted in August 2020. In December 2020, a grey literature search was performed, and supplementary searches using the search strings "Willingness to participate in clinical trials" and "neglected tropical diseases" were conducted on Google Scholar. The first 100 results by relevance were selected and screened. After the screening step, a systematic citation search ("snowballing") was conducted using the tool Connect Papers to collect all references cited in the included studies, as well as all citations received by them. All papers found were managed using the *Mendeley* software, which divided them according to the database they belonged to, removed duplicates (VRS) using Mendeley's duplicate identification strategy, and then manually.

The reproducible search strategies for all databases are presented in Supporting Information. (S3 Appendix)

### Studies selection

The scientific publications found after searching the indexed databases were screened in two stages: 1) eligibility criteria assessment through the review of titles and abstracts, and 2) eligibility criteria assessment through the full-text review. Both stages were conducted independently by two reviewers (JRF and VRS), and disagreements were resolved through discussion. The *Mendeley* software was employed for scientific publication selection.

Due to the time gap between the creation of the search strategy in 2020 and the commencement of writing this paper, we decided to update the search strategy in April 2023. We also reevaluated the study selection process with the aim of identifying articles published between August 2020 and April 2023. This step was conducted by a single reviewer, VRS.

### Data collection

Data extraction from the papers was carried out by three independent reviewers (CTM, JRF, and VRS) to avoid perception bias, using *Microsoft Excel* software and *Google Sheets* programs. The following data were extracted: author name, year of publication, country, and continent where the research was conducted, NTD, study design, major study question, population, setting, nature of the study, and nationality of the corresponding author. The nature of the study was classified as hypothetical if subjects were asked about factors related to participation in hypothetical NTD biomedical research, retrospective if subjects who had been invited to participate in biomedical NTD research in the past were asked about factors related to research participation, and prospective if subjects were asked before being invited to participate in biomedical NTD research about factors related to research participation. The nationality of the corresponding author was included as a variable to assess whether the research leader was familiar with the environment where the willingness to participate was researched. Finally, barriers and facilitators for participation, and recruitment interventions reported in the studies were extracted. A guide was adopted with the meaning of each data category to enable data extraction and avoid discrepancies between reviewers.

### Data analyses

We adapted the categories of reasons for participation and non-participation utilized by Browne et al [15]. We classified the enablers for participating into five categories: personal benefit, benefits for others, agreeable research aspects, social acceptance, and previous knowledge. We also classified the barriers to participating into five categories: physical harm, social harm, practical inconveniences, disagreement with research aspects, and personal opinions.

The four categories which we utilized to classify the recruitment interventions were adapted from UyBico et al 2007 [16] and are the following: community outreach, referrals, social marketing, and health systems.

In order to summarize results, we take into consideration the frequency of citation found in the qualitative included studies, while in included quantitative study, we considered the facilitators and barriers that were reported by the authors as statistically significant.

## Quality assessment of included studies

Two reviewers (CTM and VRS) independently performed the assessment of the methodological quality of the included studies. Disagreements were resolved through discussion. For qualitative studies, we used the CASP Qualitative Studies Checklist. Whereas for cross-sectional studies, we used the Checklist for Analytical Cross-Sectional Studies.

# Results

## Search results

The search resulted in a total of 1470 citations, out of which only ten publications met the inclusion criteria and were included in the final review. One of the publications by Pérez-Guerra C et al [17]. included two studies, one taken in 2006 and the other in 2010. Those studies have different samples and report different enablers and barriers to participating in biomedical research. Therefore, we prefer to analyze them separately in our review. Fig 1 summarizes the study selection.

## Study characteristics

Table 1 presents an overview of the included studies, which were published between 2009 and 2021. The majority of these studies were conducted in Africa (n = 8), followed by America (n = 2), and Asia (n = 1). Out of the 20 Neglected Tropical Diseases (NTDs) categorized by the WHO, the included studies focused on only five. Five studies investigated willingness to participate in research related to podoconiosis, three studies examined willingness to participate in dengue research, one study explored willingness to participate in schistosomiasis research, one study addressed willingness to participate in studies on onchocerciasis, and one study delved into willingness to participate in mycetoma research. Nearly all of the studies (n = 10) employed qualitative methods, with only one employing quantitative methods. The majority of the studies (n = 8) were carried out in rural areas, while the remainder were conducted in urban or peri-urban areas (n = 3). All of the studies had corresponding authors from the same countries where the studies were conducted. The majority of the studies (n = 8) had a nature prospective, while two had a nature hypothetical approach, and one employed a nature retrospective approach.

## Quality assessment of included studies

The ten qualitative studies were assessed based on nine quality criteria of CASP [27]. Each criterion was given either of the following three options "YES" if a criterion was met, or "CAN'T TELL" if the information present in the study is not enough to have a conclusion about the criteria, or "NO" if a criterion was not met. All studies (100%) were judged as having a clear statement of the aims of the research as well as having a qualitative methodology. Regarding the research design, six studies (54,5%) were judged as having an appropriate design to address the aims of the research, while there were also six studies (54,5%) judged as having a recruitment strategy appropriate to the aims of the research. Nine (81,8%) were judged as having

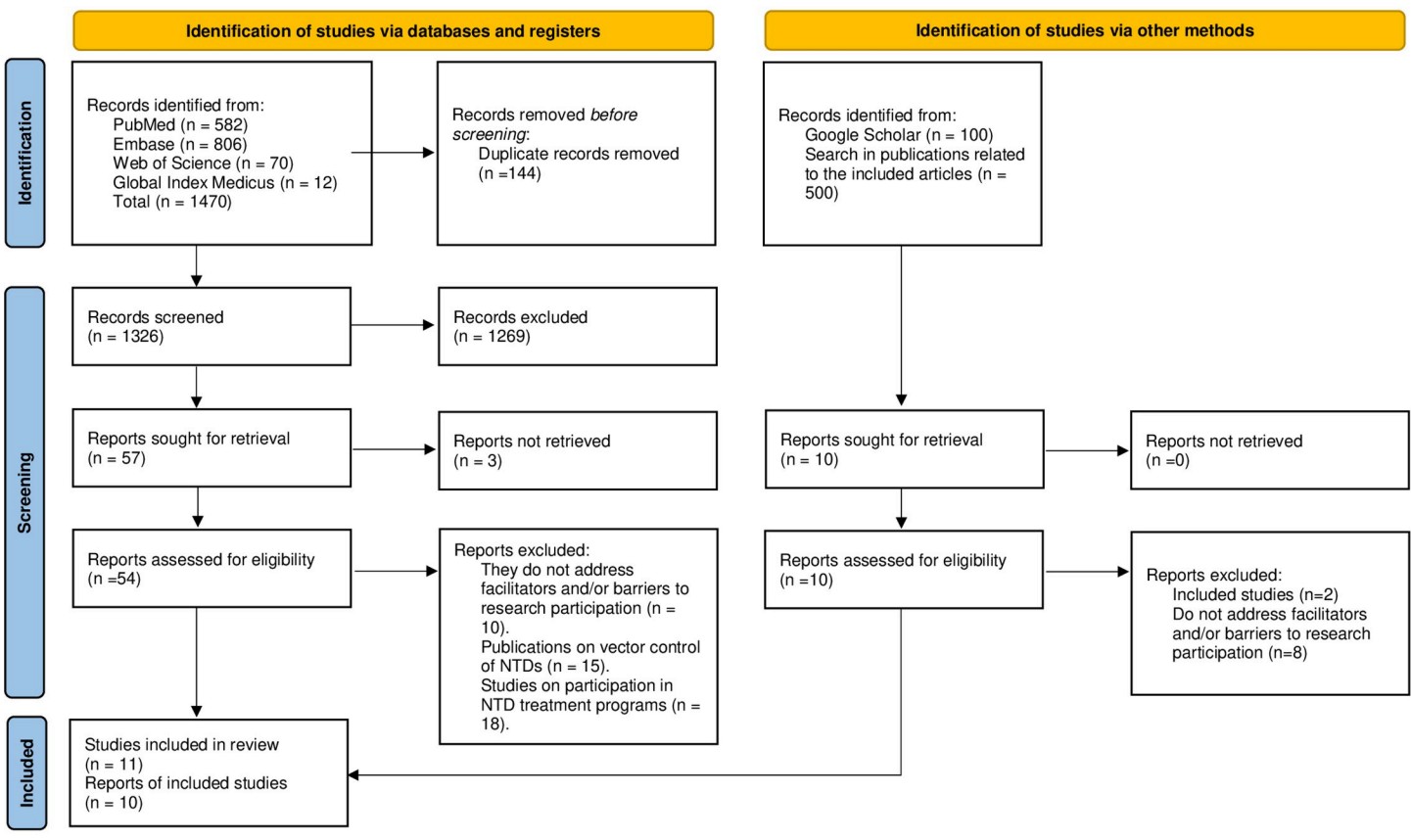

**Fig 1. PRISMA flow diagram.**

collected the data in a way that addressed the research issue, however, only two studies (18,1%) were judged as having adequately considered the relationship between the researcher and participants. There were seven studies (63,6%) that considered ethical issues, eight studies (72,7%) were judged as having done sufficiently rigorous data analysis, and another seven studies (63,63%) were judged as having clear statement findings. Only one cross-sectional study was assessed based on the Checklist for Analytical Cross-Sectional Studies [28] and matched all quality criteria of this tool.

**Factors favoring participation in biomedical research on NTD.** In this step, we did not differentiate between qualitative and quantitative studies. Table 2 summarizes the reasons favoring participation in biomedical research on NTDs and citations. The frequency was calculated considering the number of studies out of a total of eleven that cited at least one specific reason for participating in research.

The most frequently cited reason favoring participation in biomedical research on NTDS was monetary benefits or other rewards, as reported by six studies. Additionally, personal health benefits, community engagement and sensitization strategies, and comprehensive information for the study population were each cited by five studies.

**Factors serving as barriers to participation in biomedical research on NTD.** Table 3 summarizes the denial of participation in biomedical research on NTD and citations. The frequency was calculated considering the number of studies out of a total of ten that cited at least one specific reason against participating in research.

**Table 1. Characteristics of included studies.**

| 1 st author, year | Country (Continent) | Neglected Tropical Diseases | Study design | Major study question/ First aim | Study population | Setting | Nature of Study | Nationality of the corresponding author |
|---|---|---|---|---|---|---|---|---|
| Pérez-Guerra C, 2012 (Study 1) [17] | Puerto Rico (North America) | Dengue | Qualitative research: in-depth interviews (IDIs) and focus-group discussions (FGDs) | The objective of **in-depth interviews (IDIs):** to develop interview guide questions for **focus-group discussions (FGDs):** The objective of **focus-group discussions (FGDs):** to assess willingness to participate and let their children participate in a dengue vaccine trial | **in-depth interviews (IDIs):** university research staff; **focus-group discussions (FGDs):** adults with laboratory-confirmed assessed dengue and adults with no history of dengue | Urban | Prospective | Puerto Rican |
| Pérez-Guerra C, 2012 (Study 2) [17] | Puerto Rico (North America) | Dengue | Qualitative research: in-depth interviews (IDIs) and focus-group discussions (FGDs) | The objective of **in-depth interviews (IDIs):** to assess knowledge about dengue and dengue prevention, the acceptability of having a dengue vaccine and willingness to support a dengue vaccine trial; The objective of **focus-group discussions (FGDs):** to identify knowledge, attitudes and beliefs toward vaccines and vaccine trials; determine parents' willingness to allow children to participate in a dengue vaccine trial; and, identify common obstacles to recruitment | **in-depth interviews (IDIs):** university researchers, mayors, school principals, school teachers, community leaders, parents belonging to community associations; **focus-group discussions (FGDs):** children aged 9–16 y with or without a history of confirmed dengue and their parents | Urban | Prospective | Puerto Rican |
| Harapan H, 2016 [18] | Indonesia (Asia) | Dengue | Quantitative research: cross-sectional | To determine the factors that influence the willingness to participate in dengue research that would require phlebotomy procedures among participating healthy community members | Healthy community members | Urban | Hypothetical | Indonesian |
| Gebresilase T, 2017 [19] | Ethiopia (Africa) | Podoconiosis | Qualitative research: in-depth interviews (IDIs) and focus-group discussions (FGDs) | To explore barriers to getting genuine informed consent prior to enrolling participants in the genome-wide association study (GWAS) of podoconiosis in East Gojjam and East Wellega Zones of Ethiopia. | **in-depth interviews (IDIs):** Healthy community members, podoconiosis patients, field workers, researchers, religious leaders, podoconiosis administrators, local leaders; **focus-group discussions (FGDs):** Healthy community members, podoconiosis patients | Rural | Prospective | Ethiopian |

(*Continued*)

**Table 1.** (Continued)

| 1 st author, year | Country (Continent) | Neglected Tropical Diseases | Study design | Major study question/ First aim | Study population | Setting | Nature of Study | Nationality of the corresponding author |
|---|---|---|---|---|---|---|---|---|
| Akun P, 2017 [20] | Uganda (Africa) | Onchocerciasis | Qualitative research: in-depth interviews (IDIs) and focus-group discussions (FGDs) | To build dialogue between communities and the research team, help the community understand research, encourage participation, the objectives and procedures of the study, and adherence to the study interventions and schedules. | Healthy community members, health workers, and district leaders | Rural | Prospective | Ugandan |
| Negussie H, 2016 [21] | Ethiopia (Africa) | Podoconiosis | Qualitative research: in-depth interviews (IDIs) and focus-group discussions (FGDs) | To explore optimal methods to provide information about the trial and approaches to the consent process preferred by the community. Overall, suggestions were grouped into five domains: approaching patients, information provision and comprehension, decision making, constraints to participation, and ways of explaining randomization and the control group. | **in-depth interviews (IDIs):** researchers that conducted studies in the region, health professionals that work with filariasis, religious leaders, community leaders **focus-group discussions (FGDs):** community members with and without podoconiosis | Rural | Prospective | Ethiopian |
| Sanya RE, 2017 [22] | Uganda (Africa) | Schistosomiasis | Qualitative research: in-depth interviews (IDIs) and focus-group discussions (FGDs) | To investigate the perceptions of six Lake Victoria island communities of Koome, Uganda, about interventions to control Schistosoma mansoni infection and their willingness to participate in Schistosoma vaccine trials. | **in-depth interviews (IDIs):** community residents, community leaders, religious leaders, and health professionals. **focus-group discussions (FGDs):** community residents | Rural | Hypothetical | Ugandan |
| Tekola F. 2009 [23] | Ethiopia (Africa) | Podoconiosis | Qualitative research: in-depth interviews (IDIs) and focus-group discussions (FGDs) | Were assessed determinants of and approaches to gaining informed consent for biomedical research in a predominantly rural Ethiopian population, and discuss the practical ways in which we used this information in a subsequent genetic study. | **in-depth interviews (IDIs):** researchers, fieldworkers, community members, and local leaders; **focus-group discussions (FGDs):** community members | Rural | Prospective | Ethiopian |

(*Continued*)

**Table 1.** (Continued)

| 1 st author, year | Country (Continent) | Neglected Tropical Diseases | Study design | Major study question/ First aim | Study population | Setting | Nature of Study | Nationality of the corresponding author |
|---|---|---|---|---|---|---|---|---|
| Tekola F, 2009 [24] | Ethiopia (Africa) | Podoconiosis | Qualitative research: in-depth interviews (IDIs) and focus-group discussions (FGDs) | To explore the impact of social stigma on the process of obtaining consent for a study on the genetics of podoconiosis in Southern Ethiopia. | **in-depth interviews (IDIs) and focus-group discussions (FGDs):** Scientist and researchers; Field workers; coordinators, leaders, Research knowledge and managers of the MFTPA (Mossy is feet treatment and prevention association); Administrative leaders; community leaders and community residents with and without podoconiosis | Rural | Prospective | Ethiopian |
| Negash M, 2021 [25] | Ethiopia (Africa) | Podoconiosis | Qualitative research: in-depth interviews (IDIs) and focus-group discussions (FGDs) | Assess stakeholder perceptions of research, researchers and the informed consent pro-cess. | **in-depth interviews (IDIs):**review board members; health extension workers and podoconiosis focal persons (health professionals managing podoconiosis patients); podoconiosis patients; healthy community members); kebele and local religious leaders; and researchers who witnessed youth protests and property damage at the study site in the 2018 incidents. **focus-group discussions (FGDs):** zonal and regional security officials from Bahir Dar and a second for health professionals who had experience of working with the community | Rural | Prospective | Ethiopian |
| Omer R, 2021 [26] | Sudan (Africa) | Mycetoma | Qualitative research: Report on challenges and barriers encountered in the research | Report the challenges and barriers faced in the recruitment and retention of patients. | Community residents, community leaders, religious leaders, and health professionals, staff of research | Rural | Retrospective | Sudanese |

The most frequently cited reason serving as barriers to participation in biomedical research on NTDS was lack of knowledge, as reported by eight studies. Additionally, mistrust was cited by six studies.

We designed Fig 2 to make it more accessible for researchers to apply information on facili-tators and barriers in the planning of studies on Neglected Tropical Diseases (NTDs). In this figure, we outline recommended actions and those to be avoided before and during the recruitment phase, as well as after the completion of the research.

**Table 2. Factors favoring participation in biomedical research on NTD.**

| Reasons category | Frequency |
|---|---|
| *Personal benefit* | |
| ***Personal health benefits***: Possibility that the diseases are being addressed in the research. [17,19,23,24] | (5/11) |
| ***Monetary benefit or other rewards***: Participating in research is associated with receiving some kind of material or financial gain. [17,21,23–25] | (6/11) |
| ***Access to health care***: Associated with free access to medical treatment. [17,19,26] | (4/11) |
| *Benefit for others* | |
| ***Altruism***: The feeling of doing something good for the community. [17,23] | (3/11) |
| ***Community involvement***: Research brings benefits to the community of the potential participants. [21,24,26] | (3/11) |
| *Agreeable research aspects* | |
| ***Community strategies of engagement and sensibilization***: Before starting the recruitment, the research staff did events to present the research to the community and tried to integrate the local community members into the research planning and execution. [19–22,24] | (5/11) |
| ***Comprehensive information for the study population***: The research staff uses clear language that facilitates the comprehension of research by potential participants. [17,19,21,24,26] | (5/11) |
| ***Written and oral Informed Consent Form***: To avoid problems associated with illiteracy the informed consent form is obtained in written and oral aways. [19,24,25] | (3/11) |
| ***Expenditure refund***: The reimbursement for any research expenses is guaranteed for the potential participants. [17,26] | (3/11) |
| ***Guarantee of confidentiality***: The confidentiality of the information provided in the research is guaranteed. [19,22] | (2/11) |
| ***Convenience (Logistic facility)***: Taking part in the research doesn't take much time and it's accessible. [17] | (1/11) |
| ***Positive previous experience***: Potential participants had positive previous experiences with research. [19] | (1/11) |
| ***Result Availability***: Results are available at the end of the research. [19] | (1/11) |
| ***Research attitude***: There is a positive attitude by the researchers. [24] | (1/11) |
| *Social acceptance* | |
| ***Support from local leadership***: Local leadership agrees with research and encourages the community members to participate in research. [19,21,24] | (3/11) |
| ***To ensure time enough for discussion between community members and relatives***: Community members have time enough for a collective debate about participating in research. [21,22,24] | (3/11) |
| ***Integrate local community members in the research planning and execution***: Self-explanatory. [19,21,25,26] | (4/11) |
| ***Trust in researchers***: Potential participants trust the researchers and their methods. [23–25] | (3/11) |
| ***Respect for the local values (cultural and religious)***: The research staff planned the research taking into account the local values (cultural and religious) of the research setting. [24] | (1/11) |
| *Previous knowledge* | |
| ***Knowledge about research***: Potential participants have prior knowledge about research methods. [17,18] | (3/11) |
| ***Education about the disease***: Potential participants have prior knowledge about the disease covered by the research. [17,18] | (3/11) |
| ***Attitudes about the disease***: Potential participants have prior attitudes to avoid or prevent the disease covered by the research. [17,18] | (2/11) |

**Recruitment interventions.** Table 4 summarizes the types of recruitment interventions and citations. The frequency was calculated considering the number of studies out of a total of seven that cited at least one type of recruitment intervention.

The most common recruitment intervention applied in biomedical research on NTDs was contact with community leaders and organizations, as reported by five studies. Additionally, support from community health workers and referrals from friends and family were each cited by four studies.

**Table 3. Factors serving as barriers to participation in biomedical research on NTD.**

| Reasons category | Frequency |
|---|---|
| *Physical harm* | |
| **Safety concerns**: *Potential participants fear that the research will interfere with their health in some way.* [17,23] | (3/10) |
| **Invasive procedures**: *The research methods include invasive procedures.* [19,23,25] | (3/10) |
| **Worsening of the current medical condition**: *Self-explanatory.* [17] | (2/10) |
| *Social harm* | |
| **Cultural insensitivity**: *Some aspects of the research interfere with the participant's cultural norms.* [17,19,22,26] | (4/10) |
| **Lack of social support**: *People around the potential participants are not supportive of their participation in research.* [17,21] | (2/10) |
| **Stigmatization**: *Been Identified as a subject of research causes social disapproval.* [20,26] | (2/10) |
| **Confidentiality concerns**: *Potential participants fear that the confidentiality of their personal information will be violated.* [22,25] | (2/10) |
| **Gender questions**: *The research procedures conflict with local gender questions (e. g some females were not comfortable being interviewed alone with a man from research).* [19] | (1/10) |
| **Local political conflicts**: *Participating in research may be misconceived as a benefit for local politicians.* [19] | (1/10) |
| **Religious belief**: *The research procedures conflict with the religious beliefs of potential participants.* [23,26] | (2/10) |
| *Practical inconveniences* | |
| **Inconvenience**: *Participating in research brings with it a lot of troubles.* [17,21,26] | (3/10) |
| **The Large interval between sensibilization and recruitment**: *Self-explanatory.* [21] | (1/10) |
| *Disagreement with research aspects* | |
| **Mistrust**: *Distrust in the researchers.* [17,19,21–23] | (6/10) |
| **Insufficient compensation**: *Monetary or material compensation is not enough for the participant.* [19,21,22] | (3/10) |
| **Lack of clarity**: *Potential participants did not understand aspects of the research that were not adequately explained.* [19,23,24,26] | (4/10) |
| **Concerns with placebo**: *Potential participants are unwilling to receive a placebo.* [17]] | (2/10) |
| **Disagreement with research requirement**: *Potential participants disagreed with the research objective, protocols, and procedures, and therefore refused to participate.* [17,25,26] | (2/10) |
| **Communication problems between researchers and participants**: *The research staff uses a language that confuses the comprehension of research for potential participants.* [19] | (1/10) |
| **Long studies (longitudinal)**: *Potential participants consider the study duration long.* [21] | (1/10) |
| *Personal opinions /Ignorance* | |
| **Lack of knowledge**: *Potential participants have a lack of clear understanding of the research's methods.* [17,19,21–23,25,26] | (8/10) |
| **Negative previous experiences**: *Potential participants had negative previous experiences with research.* [17,19,20,24] | (4/10) |
| **"Therapeutic Misconception"**: *Potential participants confuse research objectives, protocols, and procedures with clinical treatment.* [19,21,24,25] | (4/10) |
| **Not realizing the need to participate**: *Potential participants deny the necessity to participate in research.* [19,22] | (2/10) |
| **Negative influence from media**: *Reports about research from media local resulted in an unwillingness to participate in research by potential participants.* [17] | (2/10) |
| **False stories**: *Potential participants heard false stories about research.* [21,25] | (2/10) |
| **Mistaken information about previous researches**: *Potential participants heard false stories about previous researches.* [17] | (1/10) |

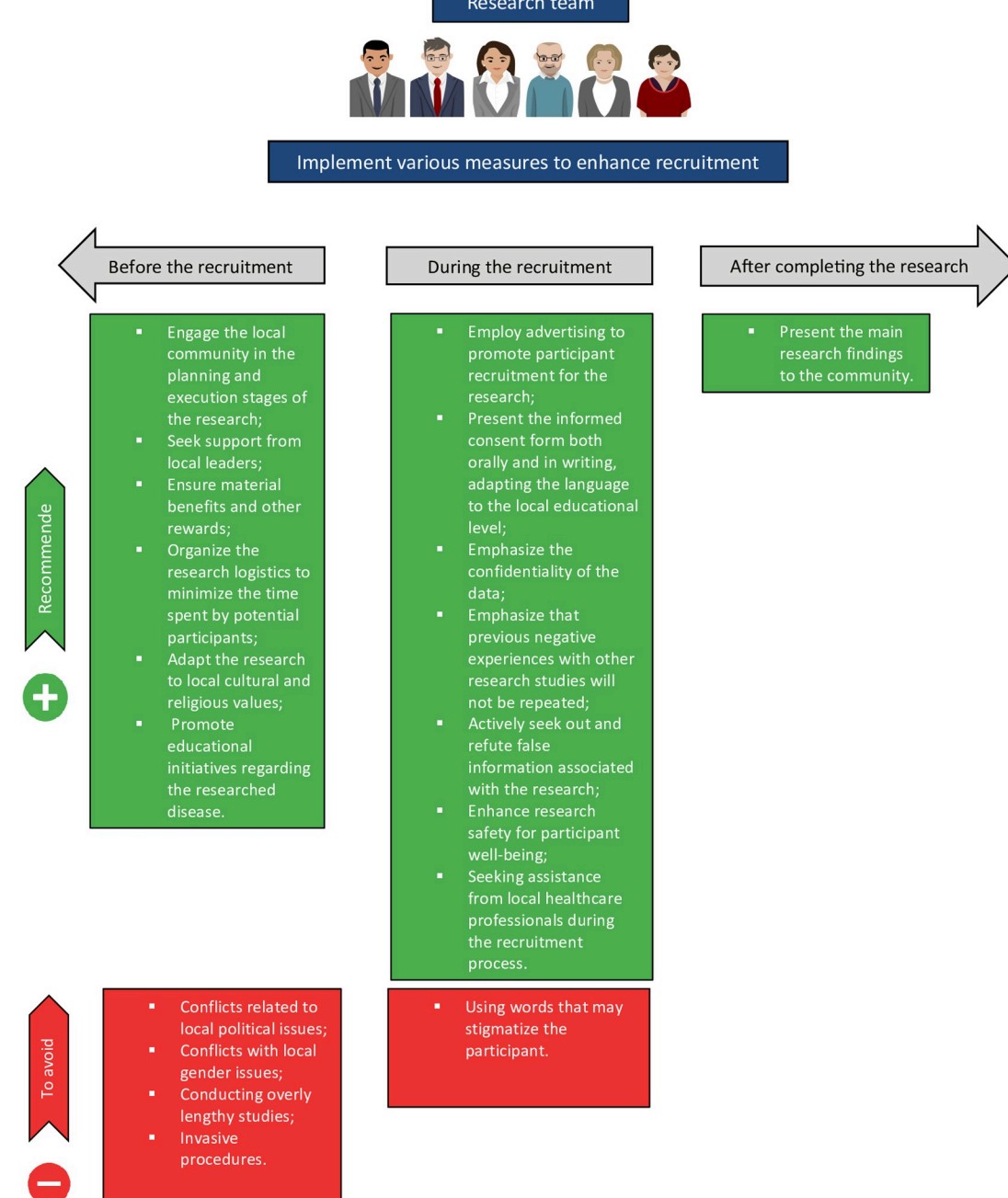

**Fig 2. Examples of potential solutions to improve recruitment rates.** Also see Tables 2 and 3 for other potential solutions. This Figure has been designed using images from 'openclipart.org'.

## Discussion

In this systematic review, we considered the frequency of citations found in the included studies to determine the main enablers and barriers to participation in biomedical research on NTDs. Monetary benefits or other rewards, personal health benefits, community engagement

**Table 4. Types of recruitment interventions.**

|  | Frequency |
|---|---|
| *Community outreach* |  |
| *Contact with community leaders and organizations. [19–21,24,26]* | (5/7) |
| *Support from community health workers. [19,20,24,26]* | (4/7) |
| *Face-to-face in the community setting. [17,20,21]* | (3/7) |
| *Going to community houses with the support of local guides or people with influence in the community. [21,26]* | (2/7) |
| *Lecture in community schools. [17,26]* | (2/7) |
| *Presentations and meetings with community residents. [24,26]* | (2/7) |
| *Referrals* |  |
| *Indication of friends and family. [19,21,22,26]* | (4/7) |
| *Indication of other study participants. [19,21,22]* | (3/7) |
| *Social marketing* |  |
| *Advertisement (Radio advertisements, Flyer, TV Advertisements, Newspaper advertisements, social media, instant messaging apps). [17,26]* | (3/7) |
| *Health system* |  |
| *Recruitment in centers of care in health. [17,24,26]* | (3/7) |

strategies and sensitization, and comprehensive information were identified as the main enablers, while lack of knowledge, and mistrust were identified as the main barriers.

After analyzing the literature on willingness to participate (WTP) in research, we have concluded that factors such as personal health benefits and altruism are commonly cited regardless of the disease being studied [29–31]. However, we have observed that depending on the field in which WTP research is being conducted, some barriers and enablers are more frequently reported. This suggests that the challenges in recruiting participants have specificities related to the context in which the research is carried out, such as the socioeconomic conditions and the disease being studied [29–38].

In a systematic review conducted by Browne et al. [15] on WTP in research involving human beings in low- and middle-income countries, a set of barriers and enablers for participation in research were identified [15]. While eleven of those barriers and enablers were not found in the studies included in our review, eleven barriers and nine enablers present in our review were not cited in the article of Browne et al [15]. Those new factors favoring participation exclusively found in our review are the following community strategies of engagement and sensibilization, comprehensive information for the study population, expenditure refund, integrate local community members in the research planning and execution, positive previous experience with researches, respect for the local values, support from local leadership, to ensure time enough for discussion between community members and relatives, and written and oral informed consent form [15]. Those new factors serving as barriers exclusively found in our review are the following: communication problems between researchers and participants, disagreement with research requirement, false stories, gender questions, local political conflicts, long studies, mistaken information about previous researches, negative influence from media, religious beliefs, the large interval between sensibilization and recruitment, and "therapeutic misconception".

This divergence between our founding and the founding of Browne et al [15] suggests that although the socioeconomic context is relevant, it alone does not fully explain the barriers and facilitators for participation in research, in such a way that intrinsic factors related to NTDs, such as the natural history of the disease, risk factors, and the epidemiological chain, can generate specific facilitators and barriers for this group of diseases

The impact of multiple factors on WTP in research can be exemplified by the role of stigmatization in the decision not to participate in a podoconiosis genetic study [23]. Without a study on WTP in research in the context of podoconiosis, it would not have been revealed that being identified as a subject of a genetic study of podoconiosis causes social disapproval and stigmatization for potential participants, leading to low recruitment rates. This illustrates the role of cultural aspects and the contribution of specificities of the disease covered by the study of WTP in research.

Our review highlighted the crucial role of social support in individuals' choice to participate in a study. Therefore, strategies such as community engagement are useful for improving recruitment rates. We also identified that lack of clarity in information has a negative impact on WTP in research. Thus, it would be important to expand the use of social media, such as journals, social networks, radio, and television, to share necessary information about the research for potential participants. In addition, we recommend the following types of recruitment interventions in the context of research on NTDS: contact with community leaders and organizations with support from community health workers, face-to-face in the community setting, and recommendations from friends and family.

In conclusion, this systematic review presents recommendations that can be applied to improve adherence in biomedical research on Neglected Tropical Diseases. A study conducted outside the context of NTDs showed that previous qualitative investigations of barriers and enablers of the recruitment process led to better recruitment rates in subsequent biomedical research [37–38]. Thus, applying the knowledge present in our review could change the probable outcome of a low recruitment rate in biomedical research carried out in the context of NTDs.

We identified some potential limitations of this systematic review. Despite the wide search conducted in the literature, only eleven studies met the inclusion criteria of our systematic review. Out of the 20 diseases classified as NTDs by WHO, we found studies on only five: dengue, mycetoma, podoconiosis, schistosomiasis, and onchocerciasis. This shows that the theme of WTP in research on NTDs is itself neglected, making it difficult to create and adopt strategies that avoid insufficient recruitment of subjects for research.

On the other hand, this systematic review has several strong points. The steps of selection, assessment, and extraction of the data were done independently by at least two authors. This reduces the chance of introducing bias, such as selection bias. Furthermore, we conducted a wide search in the literature with the assistance of a specialist (MMS). In general, this systematic review closely fulfills the methodology criterion of a systematic review, has been published previously in the protocol, and follows the recommendations for conducting a systematic review established by PRISMA and the Joanna Briggs Institute.

## Supporting information

**S1 Appendix. Protocol registered in PROSPERO.**
(PDF)

**S2 Appendix. PRISMA checklist.**
(DOCX)

**S3 Appendix. Search strategies.**
(DOCX)

## Acknowledgments

We thank Jorgana Soares, Luciano Kalabric, Rita Fernandes, and Ronald Blanton for the final review of the manuscript. We also express our gratitude to Caroline Vieira and Thiago Cerqueira for their valuable comments, which contributed to the development of Fig 2.

## Author Contributions

**Conceptualization:** Vinícius Raimundo-Silva, Mitermayer Galvão Reis.

**Data curation:** Vinícius Raimundo-Silva, Martha Silvia Martínez-Silveira.

**Formal analysis:** Vinícius Raimundo-Silva.

**Funding acquisition:** Mitermayer Galvão Reis.

**Investigation:** Vinícius Raimundo-Silva.

**Methodology:** Vinícius Raimundo-Silva, Caio Torres Marques, João Rezende Fonseca, Martha Silvia Martínez-Silveira.

**Project administration:** Vinícius Raimundo-Silva.

**Software:** Vinícius Raimundo-Silva, Martha Silvia Martínez-Silveira.

**Supervision:** Mitermayer Galvão Reis.

**Validation:** Martha Silvia Martínez-Silveira.

**Writing – original draft:** Vinícius Raimundo-Silva, Caio Torres Marques, João Rezende Fonseca, Martha Silvia Martínez-Silveira, Mitermayer Galvão Reis.

**Writing – review & editing:** Vinícius Raimundo-Silva, Caio Torres Marques, João Rezende Fonseca, Martha Silvia Martínez-Silveira, Mitermayer Galvão Reis.

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
