## [Decision Letter · Decision Letter 0]

20 Dec 2023

Dear Dr Reis,

Thank you very much for submitting your manuscript "Factors related to willingness to participate in biomedical research on neglected tropical diseases: A Systematic Review" for consideration at PLOS Neglected Tropical Diseases. As with all papers reviewed by the journal, your manuscript was reviewed by members of the editorial board and by several independent reviewers. The reviewers appreciated the attention to an important topic. Based on the reviews, we are likely to accept this manuscript for publication, providing that you modify the manuscript according to the review recommendations. 

Sincerely,

Felix Bongomin, MB ChB, MSc, MMed, FECMM

Academic Editor

Dileepa Ediriweera

Section Editor

Reviewer's Responses to Questions

**Key Review Criteria Required for Acceptance?**

**Methods**

-Are the objectives of the study clearly articulated with a clear testable hypothesis stated?

-Is the study design appropriate to address the stated objectives?

-Is the population clearly described and appropriate for the hypothesis being tested?

-Is the sample size sufficient to ensure adequate power to address the hypothesis being tested?

-Were correct statistical analysis used to support conclusions?

-Are there concerns about ethical or regulatory requirements being met?

Reviewer #1: The methods for this systematic review are sound and are in accordance with best practice (PRISMA and the PRISMA-S checklist). The authors need to be specific in the abstract and in the Search strategy and selection criteria section about which platforms they searched the database Embase on, and state that they searched the WHO Global Index Medicus, as well as which version of the Web of Science they searched (All Databases or the Core Collection?). This information needs to be provided consistently in the abstract and in the main text. 

One typo: the abstract needs to be corrected to "1470 from the databases". 

Thank you very much for including the PROSPERO protocol, the PRISMA checklist, and the reproducible search strategies in the Supporting Information, in line with best practice. 

Which software did you use to screen the papers? Mendeley?

**Results**

-Does the analysis presented match the analysis plan?

-Are the results clearly and completely presented?

-Are the figures (Tables, Images) of sufficient quality for clarity?

Reviewer #1: I have no comments to make on the results. The results are well presented and they are very clear. Thank you.

**Conclusions**

-Are the conclusions supported by the data presented?

-Are the limitations of analysis clearly described?

-Do the authors discuss how these data can be helpful to advance our understanding of the topic under study?

-Is public health relevance addressed?

Reviewer #1: The conclusions are presented in the DISCUSSION section. The conclusions are clearly supported by the data. The authors discuss how the findings could contribute to improve recruitment rates in future, thus addressing the public health relevance of their review.

**Editorial and Data Presentation Modifications?**

Reviewer #1: In terms of editorial comments: 

The authors need to check all of the references to correct the entries which are missing author names, or which have errant details. The citation style is inconsistent. Every reference needs to be checked and corrected. 

Could the authors present their recommendations about ways to improve recruitment rates in visual form to make the paper more of a pleasure to read and improve the accessibility of their findings?

**Summary and General Comments**

Reviewer #1: This systematic review is timely and unique in the literature, filling a niche and contributing to the overall literature about research participation and the state of research on NTDs overall. Thank you.

PLOS authors have the option to publish the peer review history of their article (what does this mean?). If published, this will include your full peer review and any attached files.

Reviewer #1: Yes: Elinor Harriss

Figure Files:

Data Requirements:

Please note that, as a condition of publication, PLOS' data policy requires that you make available all data used to draw the conclusions outlined in your manuscript. Data must be deposited in an appropriate repository, included within the body of the manuscript, or uploaded as supporting information. This includes all numerical values that were used to generate graphs, histograms etc.. For an example see here: http://www.plosbiology.org/article/info:doi%2F10.1371%2Fjournal.pbio.1001908#s5.

Reproducibility:

References

---

## [Editor Report · Decision Letter 1]

14 Feb 2024

Dear Dr Reis,

We are pleased to inform you that your manuscript 'Factors related to willingness to participate in biomedical research on neglected tropical diseases: A Systematic Review' has been provisionally accepted for publication in PLOS Neglected Tropical Diseases.

Best regards,

Felix Bongomin, MB ChB, MSc, MMed, FECMM

Academic Editor

Dileepa Ediriweera

Section Editor

---

## [Editor Report · Acceptance letter]

5 Mar 2024

Dear Dr Reis,

We are delighted to inform you that your manuscript, "Factors related to willingness to participate in biomedical research on neglected tropical diseases: a systematic review," has been formally accepted for publication in PLOS Neglected Tropical Diseases.

Best regards,

Shaden Kamhawi

co-Editor-in-Chief

Paul Brindley

co-Editor-in-Chief
